# LEARNABLE VISUAL WORDS FOR INTERPRETING IMAGE RECOGNITION MODELS

## ABSTRACT

To interpret deep models' predictions, attention-based visual cues are widely used in addressing *why* deep models make such predictions. Beyond that, the current research community becomes more interested in reasoning *how* deep models make predictions, where some prototype-based methods employ interpretable representations with their corresponding visual cues to reveal the black-box mechanism of deep model behaviors. However, these pioneering attempts only either learn the category-specific prototypes with their generalization ability deterioration or demonstrate several illustrative examples without a quantitative evaluation of visual-based interpretability narrowing their practical usage. In this paper, we revisit the concept of visual words and propose the Learnable Visual Words (LVW) to interpret the model prediction behaviors with two novel modules: semantic visual words learning and dual fidelity preservation. The semantic visual words learning relaxes the category-specific constraint, enabling the generic visual words shared across multiple categories. Beyond employing the visual words for prediction to align visual words with the base model, our dual fidelity preservation also includes the attention-guided semantic alignment that encourages the learned visual words to focus on the same conceptual regions for prediction. Experiments on six visual benchmarks demonstrate the superior effectiveness of our proposed LVW in both accuracy and interpretation fidelity over the state-of-the-art methods. Moreover, we elaborate on various in-depth analyses to further explore the learned visual words and the generalizability of our method for unseen categories.[1]

## 1 INTRODUCTION

Model interpretation aims to explain the black-box base model in a semantically understandable way and preserve high fidelity with model outputs (Li et al., 2021; Molnar et al., 2020; Zhang et al., 2021b). Although interpretative models are not designed to pursue higher performance than the base model, they are of great importance, especially in the deep learning era. With human-understandable explanations, model designers can diagnose errors and potential biases embedded in deep models; model users are confident and relaxed to rely on model predictions. This trust is built on the assumption that the interpretation model is faithful to the original deep networks. For tabular data, the raw features with physical meanings are used to interpret the model prediction. LIME (Ribeiro et al., 2016) and SHAP (Lundberg & Lee, 2017) are two representative explainable models, which learn linear models to locally fit the base model's outputs with sample perturbations. For visual data, attention-based visual cues are widely adopted in interpreting deep models' predictions (Correia & Colombini, 2021). Grad-Cam (Selvaraju et al., 2017; 2020) is a popular technique for visualizing the region where a convolutional neural network model looks at. It uses the category-specific gradient information flowing into the final convolutional layer of a CNN to produce a coarse localization attention map of the important regions/pixels in the image. Following this direction, several studies further employ weakly supervised information directly on attention maps during the training stage to reduce the attention bias (Li et al., 2018; Srinivas & Fleuret, 2019; Chaudhari et al., 2021).

Beyond the above studies addressing *why* deep models make their predictions at the generally coarse level, some pioneering attempts have been taken to reason *how* deep models make predictions at the fine-grained level. ProtoPNet (Chen et al., 2019) learns a predetermined number of prototypes per

---

[1]Our code is available at *https://github.com/LearnableVW/Learnable-Visual-Words*.

category and proposes a prototypical part network with a hidden layer of prototypes representing the activated patterns. By learning these prototype-based interpretable representations, ProtoPNet explains the model reasoning process by dissecting the query image into several prototypical parts and interpreting these prototypes with training images in the same category. Later, ProtoTree (Nauta et al., 2021) and ProtoPShare (Rymarczyk et al., 2021b) extend ProtoPNet by decision trees and prototype pruning to achieve global interpretation and reduce the model complexity, respectively. Unfortunately, these studies either learn the category-specific prototypes that deteriorate their generalizing capacities or only demonstrate several illustrative examples without a quantitative evaluation of visual-based interpretability (Figure 2 demonstrates their discovered meaningless or irrelevant prototypes despite their high recognition accuracy). Moreover, they only evaluate on two object-cropped visual datasets and leave their performance on other general visual datasets and samples from unseen categories on the shelf. It is crucial to note that an interpretation model should be evaluated by whether the interpretation model is loyal to the base model in all aspects of the outputs, rather than solely focusing on if it achieves higher accuracy or delivers human cognition-consistent explanations.

**Contributions**. The process of ProtoPNet recalls us of the bags-of-visual words (Csurka et al., 2004; Sivic & Zisserman, 2003), a popular image representation technique before the deep learning era. It treats an image as a document and visual words are defined by the keypoints/descriptors/patches that are used to construct vocabularies. Then the image can be represented as a histogram over the occurrences of these visual words. It is worth noting that these visual words with semantic meanings across different images can also be used for model interpretation, which is similar to the prototypes in ProtoPNet. The major difference lies in that conventional visual words are usually hand-crafted, pre-defined, and independent of the downstream learning tasks. In light of this, we propose Learnable Visual Words (LVW) to overcome the aforementioned drawbacks of the prototype-based interpretative methods. Technically, our model consists of two modules, semantic visual words learning, and dual fidelity preservation. The semantic visual words learning relaxes the category-specific constraint, enabling the generic visual words to be shared across different categories, while the dual fidelity preservation encourages the learned visual words to behave similarly to the base model in both prediction and model attention. This novel attention fidelity ensures that the learned visual words attend to the same areas of a sample image when the base deep network makes its predictions. Our major contributions are summarized as follows:

- In the semantic visual words learning, we relax its category-specific constraint and further simplify ProtoPNet to achieve cross-category visual words and increase the generalization of model interpretation, rather than adding any new terms.

- In the dual fidelity preservation, we encourage the learned visual words to preserve high fidelity with the base model in terms of both prediction and model attention. This dual fidelity helps our model identify an interpretation that loyally presents the base network. Additionally, we further design a measurement to quantitatively evaluate the visual-based interpretation.

- We demonstrate the superior effectiveness of our model on six visual benchmarks over the state-of-the-art prototype-based methods in both accuracy and visual-based interpretation and explore the generalization of our model by interpreting unseen categories.

## 2 RELATED WORK

**Visual Image Understanding**. Research efforts in interpretable explanations of a Convolutional Neural Network (CNN) can be generally divided into *posthoc* and *self-interpretable* genres. Posthoc methods attempt to build an extra explainer for the pre-trained black-box model to interpret its prediction. Approaches including saliency visualization based on backpropagation (Springenberg et al., 2014; Zhou et al., 2016; Zhang et al., 2018; Bach et al., 2015; Sundararajan et al., 2017; Shrikumar et al., 2017; Smilkov et al., 2017; Fong & Vedaldi, 2017; Rebuffi et al., 2020) and activation maximization caused by perturbation (Simonyan et al., 2013; Zeiler & Fergus, 2014; Petsiuk et al., 2018; Fong et al., 2019; Kapishnikov et al., 2019; Dabkowski & Gal, 2017; Ancona et al., 2017) to identify the most influential parts for the black-box model's prediction. However, visualizing the salient areas does not explain **how** the black box makes such decisions. Other posthoc methods (Ghorbani et al., 2019; Zhang et al., 2021a; Olah et al., 2018; Akula et al., 2020; Yeh et al., 2020; Koh et al., 2020; Kim et al., 2018; Chen et al., 2020) obtain interpretable concept activation vectors from pre-segmented feature maps and interpret the CNN model with these concepts.

Alternatively, self-interpretable methods aim to directly learn explanatory representations during training, instead of disentangling the pre-trained black box. ProtoPNet (Chen et al., 2019) introduces a case-based study that interprets the reasoning process of the black-box model with category-specific prototypes associated with image patches from training data, which answers how deep models make predictions by linking the test image with interpretable prototypes. Recent studies (Rymarczyk et al., 2021a; Donnelly et al., 2021) extend ProtoPNet in various directions. For instance, ProtoTree (Nauta et al., 2021) combines prototype learning with decision trees, resulting in an interpretable decision path consisting of prototypes. Similarly, HPnet (Hase et al., 2019) hierarchically organizes prototypes to classify objects at every level in a predefined class taxonomy. ProtoPShare(Rymarczyk et al., 2021b) further groups similar prototypes discovered by a pre-trained ProtoPNet model with a data-dependent merge-pruning strategy. Meanwhile, TesNet (Wang et al., 2021) extracts prototypes from class-specific embedding subspaces defined on the Grassmann Manifold. Lately, Deformable ProtoPNet(Dai et al., 2017) explicitly captures pose variations and context by integrating deformable convolutional networks in their model.

**Interpretation Model**. A great number of researchers are dedicated to the attempts of opening the black-box deep networks. Early works on model interpretation try to learn a set of rules or decision tree globally as the interpretation for a pre-trained deep model. Some methods (Quinlan, 2014; Odajima et al., 2008; Nayak, 2009; Krishnan et al., 1999; Pedapati et al., 2020) extract a global set of rules or a decision tree from the training samples generated by the base network. Later works seek to find local logic rules for one specific sample or a small group of samples. Dhurandhar et al. (2018) construct such rules by identifying the import features, which is represented by a sparse permutation of input features that is sufficient to produce the same prediction as the base model, while Wang et al. (2018) propose to look for critical data routing paths of the network for each input by assigning sparse non-negative weights to each channel on each layer. On the other hand, attribution-based explanation methods interpret deep networks by assigning important scores to the input features. Most of these attribution methods can be separated into two groups: gradient-based attribution and model agnostic attribution. Baehrens et al. (2010) explain the classification decision of $k$-NN and SVM with gradient-based attribution, while other methods (Springenberg et al., 2014; Bach et al., 2015; Sundararajan et al., 2017; Srinivas & Fleuret, 2019) calculate gradient-related saliency maps as attributions. Alternatively, model agnostic methods like LIME (Ribeiro et al., 2016) and MAPLE (Plumb et al., 2018) interpret individual model predictions based on locally approximating the model around a given prediction with a local linear explanation model. Other methods (Lundberg & Lee, 2017; Ancona et al., 2019) adopt Shapley values for locally measuring the attribution of input features. Lately, SpRAy (Guo et al., 2019) and MAME (Hatt et al., 2019) combine the individual attributions obtained from local methods to generate a global attribution-based explanation.

## 3 METHOD

### 3.1 FRAMEWORK OVERVIEW

We introduce the framework of our Learnable Visual Words (LVW) model to extract meaningful semantic information for interpreting the base CNN models in Figure 1, which consists of visual words learning and dual fidelity preservation. The visual words learning module aims to extract semantics shared across all categories, while the dual fidelity preservation module guides these learned visual words to focus on the regions of the image that base CNN models attend to when making predictions and also preserves the similar predictive ability of the base model. In order to learn cross-class visual words, our semantic learning module simplifies ProtoPNet (Chen et al., 2019) by removing its separation loss, which is used for finding category-specific prototypes for each category. We keep ProtoPNet's clustering loss so that the learned visual words are closely related to the training images, and at the same time, contain various shared semantic information from all classes. The dual fidelity preservation module assures the learned visual words preserve high fidelity with the base model in terms of both prediction and attention in a similarity-based approach. For each image sample, the similarity scores between one visual word and all patches of the sample image's backbone convolution output are calculated, resulting in a similarity heatmap that represents how strong this visual word matches different parts of the image, the same as ProtoPNet. The maximum value of the similarity heatmap for each visual word is then used for making the final prediction by the fully connected layer with cross-entropy loss. To achieve interpretability for the prediction made by the CNN backbone, the attention-guided alignment module combines the similarity heatmaps of

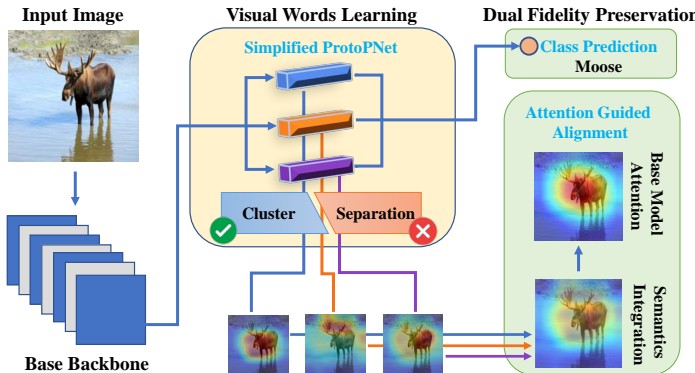

Figure 1: Illustration of our proposed Learnable Visual Words (LVW) model for image recognition interpretation. Visual Words Learning module simplifies ProtoPNet and learns cross-category semantic visual words to describe the samples; the Dual Fidelity Preservation module enhances the learned visual words to preserve high fidelity with the base model in terms of both prediction and attention.

the sample image's top $k$ visual words with max-pooling and guides the pooled heatmap with the base CNN model's class-level attention. With attention alignment, the learned visual words focus on the same areas used by the base model for making predictions, preserving the base model's attention.

## 3.2 LEARNABLE VISUAL WORDS FOR INTERPRETABLE MODEL

Since our model simplifies ProtoPNet in visual words learning, we first introduce the base backbone model and ProtoPNet to provide enough preliminaries and then elaborate on our innovations.

**Preliminaries**. Let the base backbone model (e.g., ResNet (He et al., 2016), VGG (Simonyan & Zisserman, 2014)) be trained in a supervised setting with $N$ labeled training samples $\mathcal{X} = \{(x_i, y_i)\}, 1 \leq i \leq N$ associated with $C$ different categories. ProtoPNet (Chen et al., 2019), a prototype-based explanation method for a CNN model, consists of a convolutional layer $f$, a prototype layer $g$ and a prediction layer $h$. It first extracts the feature map $Z_i = f(x_i)$ with a dimension of $H \times W \times D$ from the last convolutional layer's output of the base model. Then ProtoPNet learns total $M$ prototypes or visual words $\mathcal{V} = \{v_j\}|_{j=1}^{M}$ with the shape of $H' \times W' \times D$, $H' < H$ and $W' < W$. Their prototype layer $g$ computes an activation map consisting of the inverted $L_2$-distance between one learned visual word $v_j$ and all $H_1 \times W_1 \times D$-shape patches of $Z$. Then the activation map is resized back to the original image size, resulting in an activation heatmap $A_i^j$ to represent how strong $v_j$ matches each part of the input image $x_i$. ProtoPNet takes the maximum value in $A_i^j$, i.e., $\max(A_i^j)$ as a similarity score between the visual word $v_j$ and the input image. Later, a fully-connected layer $h$ makes the prediction based on similarity scores $g \circ f(x_i) = [\max(A_i^1), ..., \max(A_i^M)]$ associated with all $M$ learned visual words. Note that the base model's attention map of $\bar{A}_i$ for sample image $x_i$ can be calculated with existing attention method like Grad-Cam (Selvaraju et al., 2017; 2020) or GAIN (Li et al., 2018). By aligning the base model's attention $\bar{A}_i$ and the attention maps of the $k$ top visual words that are most semantically closest to $x_i$, our proposed model aims to learn visual words that keep high fidelity with the base model in terms of both prediction and attention, providing a more reasonable explanation for understanding its reasoning of making predictions.

**Visual Words Learning**. Our visual words learning module is a simplified version of ProtoPNet where we keep its clustering loss but discard the separation loss. The clustering loss encourages all training images to be close to some learned visual words; on the other hand, removing the separation loss relaxes the category-specific constraint and allows the learned visual words to be shared across categories. Mathematically, the cluster loss is defined as:

$$\mathcal{L}_c = \frac{1}{N} \sum_{i=1}^{N} \min_{v_j} \min_{z \in Z_i} ||z - v_j||^2, \tag{1}$$

where $z$ is a single patch of shape $H' \times W' \times D$ in $Z_i$.

**Attention Guided Semantic Alignment**. Beyond employing the visual words for prediction, our dual fidelity preservation also includes the attention-guided semantic alignment, which requests the learned visual words to focus on the same region as the base CNN model. To achieve this goal of attention fidelity, the alignment module combines the activation heatmaps of the top $k$ visual words with the highest similarity after max-pooling, resulting in a single activation heatmap $A^k$ that

represents how strong these $k$ visual words are present in the input image $x$. The attention alignment abides by the loss term as follows:

$$\mathcal{L}_a = \frac{1}{N} \sum_{i=1}^{N} ||\bar{A}_i - \hat{A}_i^k||^2,$$

(2)

where $\bar{A}_i$ is the input class-level attention of the base model on input image $x_i$, and $\hat{A}_i^k$ is the concatenated activation heatmap from the top $k$ visual words that are semantically closest to $x_i$.

### 3.3 TRAINING PROTOCOL AND OBJECTIVE FUNCTIONS

We adopt the same end-to-end training procedure as ProtoPNet(Chen et al., 2019) with three stages: (1) optimization before the last layer; (2) projection of visual words; (3) optimization of the last layer.

**Optimization Before the Last Layer**. In this stage, we fix the fully connected layer $h$ and train the convolutional layers $f$ and visual words layer $g$. For each category, the connections between $M/c$ visual words are set to 1 and the rest connections are set to -0.5, encouraging the model to learn some visual words that are closely related to this category. Different from ProtoPNet, we remove the separation loss so that the learning module does not force the learned visual words to be category-specific. The overall objective function for this stage is:

$$\min_{f,g} \frac{1}{N} \sum_{i=1}^{N} \mathcal{L}_{cls}(h \circ g \circ f(x_i)) + \alpha \mathcal{L}_c + \beta \mathcal{L}_a,$$

(3)

where $g \circ f(x_i)$ is a vector containing similarity scores between $M$ visual words to $x_i$, $\alpha$ and $\beta$ are two trade-off parameters.

**Projection of Visual Words**. During this stage, we project each visual word $v_j$ onto the nearest patch across all training images to further ensure that the learned visual words contain various semantic information from the training data. The project can be described as follows:

$$v_j \Leftarrow \min_{z \in \mathcal{Z}} ||z - v_j||^2,$$

(4)

where $\mathcal{Z}$ is the collection of all latent patches of shape $H' \times W' \times D$ contained in $\mathcal{X}$.

**Optimization of the Last Layer**. In this stage, we only optimize the fully connected layer $h$ with fixed convolutional layers $f$ and visual words layer $g$. This training stage uncovers the semantic associations of each learned visual word across all categories. Following ProtoPNet, we also add $L_1$-regularization during this training stage and the objective function is:

$$\min_{h} \frac{1}{N} \sum_{i=1}^{N} \mathcal{L}_{cls}(h \circ g \circ f(x_i)) + \gamma \sum_{w \in W_h} |w|,$$

(5)

where $W_h$ is the collection of all weights in the fully connected layer and $\gamma$ is the trade-off parameter.

## 4 EXPERIMENTS

We demonstrate the performance of our learnable visual words in two aspects: classification ability and attention-based model interpretability. We first introduce the experimental setup, then report the algorithmic performance with extended examples, and finally provide various in-depth analyses.

### 4.1 EXPERIMENTAL SETUP

**Datasets**. We include six datasets for performance evaluation including *STL10* (Coates et al., 2011), *Oxford 102 Flower* (Nilsback & Zisserman, 2008), *Oxford-IIIT Pets* (Parkhi et al., 2012), *Stanford Dogs* (Khosla et al., 2011), *Food-101* dataset (Bossard et al., 2014), and *AwA2* (Xian et al., 2019). Their detailed descriptions can be found in Appendix A.1.

Table 1: Performance of different interpretative models by accuracy and IoU coverage on six datasets

| Method | STL10 | | Oxford Flower | | Oxford-IIIT Pets | | Stanford-Dogs | | Food-101 | | AWA2 | |
|---|---|---|---|---|---|---|---|---|---|---|---|---|
| | Acc | IoU | Acc | IoU | Acc | IoU | Acc | IoU | Acc | IoU | Acc | IoU |
| ResNet-34 (He et al., 2016) | 87.40 | 1.00 | 98.17 | 1.00 | 92.28 | 1.00 | 79.03 | 1.00 | 81.60 | 1.00 | 93.66 | 1.00 |
| ProtoPNet (Chen et al., 2019) | 86.33 | 0.2199 | 86.33 | 0.3081 | 82.12 | 0.2190 | 69.27 | 0.4337 | 74.93 | 0.4135 | 89.18 | 0.2294 |
| ProtoTree (Nauta et al., 2021) | 89.58 | 0.3228 | 30.44 | 0.3418 | 62.41 | 0.5113 | 41.00 | 0.4267 | 36.37 | 0.4173 | 86.54 | 0.6197 |
| ProtoPShare (Rymarczyk et al., 2021b) | 87.04 | 0.2722 | 87.04 | 0.2958 | 74.70 | 0.3467 | 68.70 | 0.4585 | 72.37 | 0.4826 | 84.10 | 0.2571 |
| TesNet (Wang et al., 2021) | 87.66 | 0.5444 | 91.69 | 0.6708 | 85.53 | 0.7038 | **75.47** | 0.6387 | 75.94 | 0.5817 | 90.82 | 0.6665 |
| Ours w/o Atte. Align. | 89.55 | 0.2209 | 89.85 | 0.3722 | 85.47 | 0.2583 | 70.72 | 0.2748 | 75.03 | 0.3128 | **91.16** | 0.2536 |
| Ours | **89.75** | **0.6465** | **91.81** | **0.7967** | **86.20** | **0.8086** | 73.81 | **0.7932** | **76.33** | **0.6969** | 90.85 | **0.7781** |

**Baseline Models**. We compare our model to four baseline methods,[2] including ProtoPNet (Chen et al., 2019), ProtoTree (Nauta et al., 2021), ProtoPShare (Rymarczyk et al., 2021b) and TesNet (Wang et al., 2021). We choose ResNet-34 (He et al., 2016) as the base CNN backbone for all models including our model. For ProtoPNet, ProtoPShare, and TesNet, 10 prototypes are selected per category. For ProtoTree, we train a tree of depth 10 so that each sample image will be associated with 10 prototypes in the test stage. For our method, we choose $M = 5 \times C$ visual words, where $C$ is the class number.

**Evaluation**. Recognition accuracy is used to evaluate the predictive ability of each model. An interpretation model should not only provide human-understandable explanations of the base model but also keep loyal to the base model, so that the model understanding can be used for downstream tasks like model diagnosing and debugging. Thus we also propose an Intersection over Union (IoU) coverage metric as an objective measurement for the model's interpretability as follows:

$$\text{IoU}_{q,m} = \frac{\mathcal{M}_q(\bar{A}) \cap \mathcal{M}_q(A^k)}{\mathcal{M}_q(\bar{A}) \cup \mathcal{M}_q(A^k)}, \tag{6}$$

where $\bar{A}$ is the class-level attention from the base model, and $A^k$ is the combined attention acquired from the $m$ most related visual words. Here, $\mathcal{M}_q$ means the binary masking function that masks out all the values that smaller than the $q$-th quantile of the attention map as 0 and keeps the rest values as 1. Here we set $k = 10$ and $q = 50$ by default. Essentially, the designed IoU metric represents the amount of overlap between each sample's base model attention and the top $k$ visual words' attention. Thus, we use the average IoU across all test samples to measure the model's ability in explaining the decision made by the base CNN model. As obtaining the class-level attention maps from the base network only requires class labels, our model does not require any extra information for training. Also, our model does not have access to any attention maps of the testing samples. Thus, we believe evaluating our model against the baselines with this proposed IOU metric is a fair comparison.

**Visualization Protocol**. We adopt a similar protocol of visualizing learned visual words as ProtoPNet (Chen et al., 2019). For visual word $v_j$ globally, we identify the training image $x_i$ whose convolutional output $Z_i$ has the highest activation of $v_j$. The global visualization of $v_j$ is generated by overlaying the up-sampled activation heatmap $A_j$ on $x_i$. For any test image $x'$ locally, the activation heatmap $A_j$ of visual word $v_j$ on $x'$ represents the presence of $v_j$ on $x'$.

## 4.2 ALGORITHMIC PERFORMANCE

Here we comprehensively evaluate our learnable visual words model and three prototype-based interpretable recognition models with classification accuracy as well as the proposed IoU coverage metric. We also visually demonstrate a case study on *Food-101* dataset.

**Quantitative Evaluation**. Table 1 shows the recognition accuracy and the proposed IoU coverage of all models on six datasets, in which we highlight the best results in bold. In addition to our complete model, we also report the performance of our model without attention-guided alignment in order to check the efficacy of this module. In general, our proposed model achieves comparable recognition accuracy on these six datasets with the base ResNet-34 model. Our model sacrifices at most 6.36% accuracy loss on *Oxford Flower* (Nilsback & Zisserman, 2008), while, to our surprise, improving ResNet-34 by 2.35% on *STL10* (Coates et al., 2011) dataset. Compared with the three baseline models, our full model attains the best recognition accuracy in 5 out of 6 datasets, only second to our model without attention alignment on *AWA2* (Xian et al., 2019), which boosts prediction accuracy over

---

[2]We notice a new variants of ProtoPNet, Deformable ProtoPNet (Donnelly et al., 2021). We do not include this method for comparison, since this method uses Deformable Convolutional Networks(Dai et al., 2017), which employ a different architecture from the base CNN models used in our experiments.

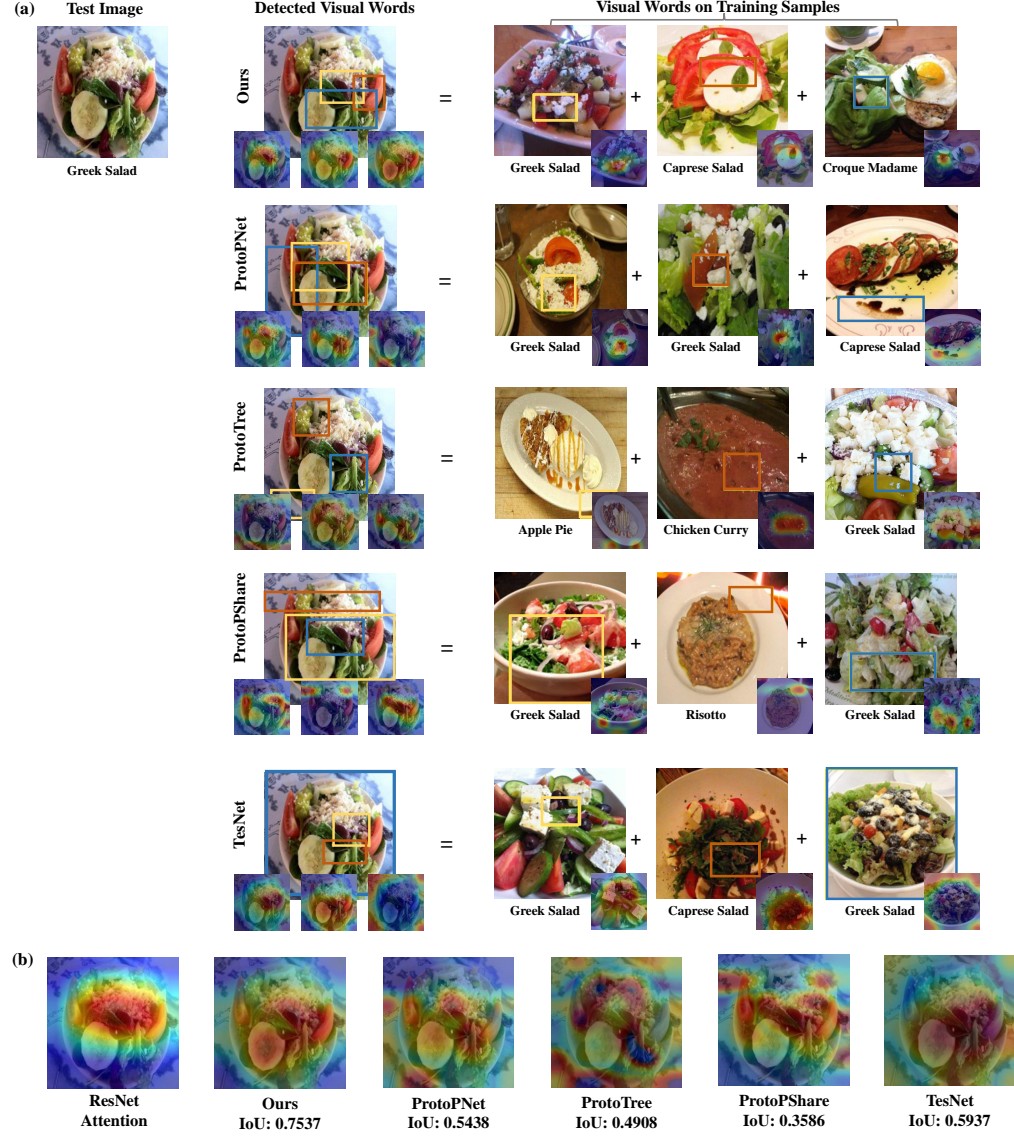

Figure 2: Visual demonstration of our model and other prototype-based methods for the same base model (ResNet-34) interpretation on *Food-101*. (a) shows the detected visual words or prototypes on the test image and corresponding training samples, where the bounding boxes are generated according to the activated regions of visual words or prototypes. We use the same technique in ProtoPNet Chen et al. (2019) for bounding box generation. (b) shows the ResNet attention map and the integrated attention maps of visual words or prototypes with their IoU values.

ProtoPNet (Chen et al., 2019) only by removing the class-specific constraint. We assert that our model preserves, and possibly improves, the predictive ability of baseline ResNet-34. However, accuracy alone cannot properly estimate the visual-based interpretability of different models (We will provide more evidence in the next Visual Demonstration paragraph). Therefore, we also evaluate all methods with our proposed IoU coverage metric to check if the explanation keeps the original meaning of the base network's reasoning process. Our model exceeds all baseline models by a relatively large margin in terms of IoU coverage. At a minimum, our model outperforms TesNet(Wang et al., 2021), which is the best-performing baseline model in terms of IOU, by at least 0.1 on all datasets, while our model significantly outperforms the rest of the baseline models, especially on datasets like *Oxford Flowers* and *STL10* (Coates et al., 2011), where our model's IOU can be twice as the IoU of the second-best baseline method. These results suggest that our learned visual words preserve the base model's

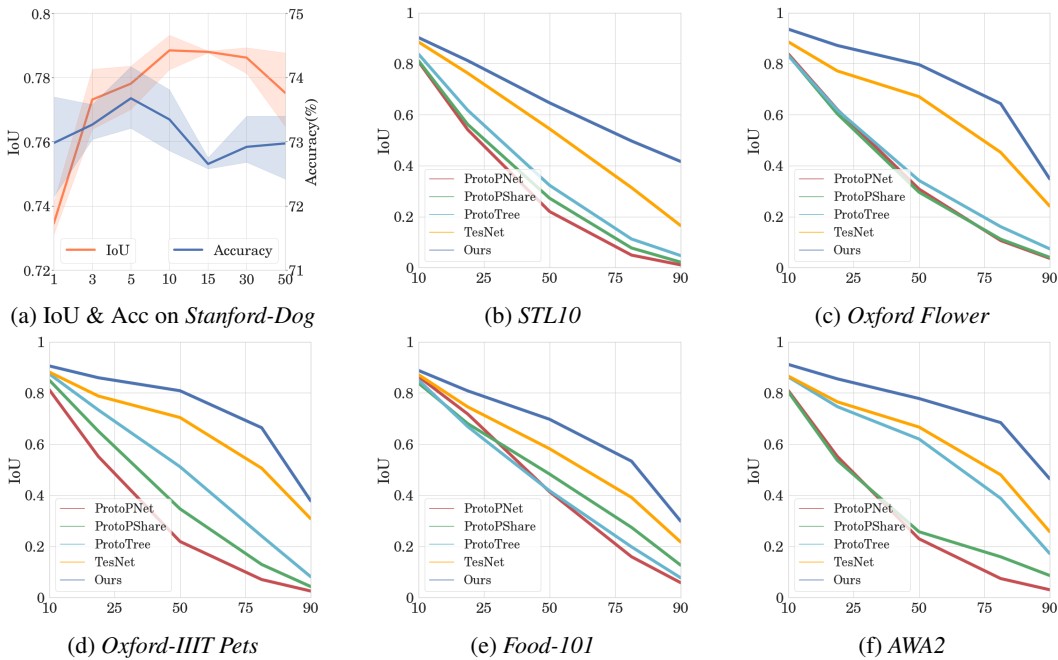

Figure 3: (a) Performance of our method with different numbers of top visual words $k$. (b-f) IoU coverage for ours and baseline models with different quantile thresholds on *STL10*, *Oxford Flower*, *Oxford-IIIT Pets*, *Food-101* and *AWA2*.

attention and provide an interpretation that is loyal to the reasoning of the base model. Notice that our model without attention alignment outperforms ProtoPNet on all six datasets in terms of accuracy, which verifies our assumption that different categories might share common visual semantics. On the other hand, the IoU coverage of the model without attention alignment is worse than ProtoPNet on datasets like *Stanford-Dogs* (Khosla et al., 2011) and *Food-101*(Bossard et al., 2014) despite the improved accuracy, indicating that only relaxing the category-specific constraint does not enhance its interpretability, where the cross-category assumption might not be suitable for fine-grained classes. Also notice that the comparison between ProtoPNet and our model without attention alignment, which is essentially ProtoPNet after removing the separation loss, illustrates that the separation loss does not affect the accuracy or IOU in any deterministic way.

**Visual Demonstration**. We compare our model with the four prototype-based methods visually with a case-study example in Figure 2. We choose one test image from *Food-101* Bossard et al. (2014) datasets and select the three most related visual words or prototypes extracted by each method. We visualize them in the test images as well as their corresponding projected training images with the same protocol as ProtoPNet(Chen et al., 2019) in Figure 2(a). Our method detects the visual words semantically corresponding to "cheese crumbs," "tomatoes," and "green vegetables" across three training categories. On the other hand, although the baseline models can also uncover prototypical patches from the sample images, the detected prototypes are more likely to focus on meaningless or irrelevant semantics which provides little explanation for the prediction. For example, all four baselines identify prototypes related to the plate located on the edge of the test image. Those prototypes might help classify this test image as "greek salad," but they are unable to interpret this decision. We conjure that without the guidance of base model attention, these prototype-based methods rely on those prototypes of the base model's attention area to make predictions, especially on the uncropped image data. In Figure 2(b), both the IoU coverage values and the combined attention maps illustrate that the ResNet attention is well covered by our visual words, while the baseline methods focus more on the edge area of the test image. From this visual demonstration, we can observe that our learnable visual words provide a more reasonable interpretation of the reasoning process of the base model than competitive methods.

**Hyperparameter Analysis**. We explore our model's sensitivity to hyperparameters on different datasets. One key hyperparameter in our model is the number of top visual words $k$ used in attention-guided semantic alignment. We choose different values of $k$ from 1 to 50 and conduct experiments

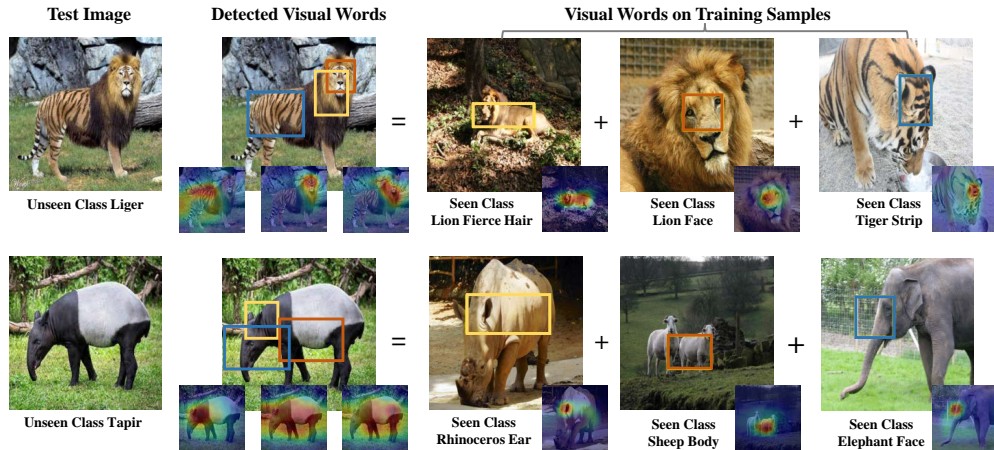

Figure 4: Illustrative examples of our learnable visual words trained on *AWA2* for interpreting image samples of Liger and Tapir. Their categories are unobserved in the training set.

on *Stanford-Dogs* dataset. We train our model for each choice of $k$, and randomly select 3 different subsets of the original test set, each containing 50% of total test images, to check our model robustness against different test data. As shown in Figure 3(a), prediction accuracy does not fluctuate for various choices of $k$. The IoU drops for smaller $k$ like 1 or 3 since we only encourage very few visual words to focus on the class-level attention. For each $k$, the error between three random test sets is insignificant, considering our large IoU improvement. Moreover, the choice of quantile threshold $q$ in our designed measurement could also significantly affect the final results of IoU during the testing stage. To evaluate the robustness and generalizing ability, we plot the IoU coverage for 5 different quantile thresholds on the rest of the five datasets. Our model consistently outperforms the four baseline models on all benchmarks as shown in Figure 3(b-f). For $q = 90$, the attention of baseline models barely overlaps with the base model attention except for TesNet(Wang et al., 2021), while our method still maintains at least 0.2 IoU coverage, which evinces our proposed method uncovers the most attended semantics in the base model.

**Interpreting Samples From Unseen Categories**. In light of the visual words across different categories, we extend our method to those categories that potentially share semantics with existing training categories, but are unseen by the base model. In Figure 4, we choose "Liger" and "Tapir" as the unseen categories and test them on our model trained on *AWA2* dataset. Ligers are zoo-bred hybrid offsprings of a male lion and a female tiger, which possess features of both parents. Tapirs are rare creatures that are often confused with pigs or anteaters, but their closest living relatives are actually rhinoceros and horses. As expected, our model successfully uncovers several shared semantics between the unseen classes with classes that exist in the training data. The face, fierce hair, and strip pattern on the body of this liger are tracked back to the learned visual words from the seen lion and tiger categories in the first row of Figure 4. Although our model does not predict this unseen image as a liger, it does help us to better understand this image: an animal with the face and fierce hair of a lion, as well as a tiger's stripe on its body. Similarly, our model can interpret this tapir as a creature, which has the ear of a rhinoceros, the body of a sheep, and the face of an elephant.

## 5 CONCLUSION

To sum up, we presented Learnable Visual Words (LVW), a novel approach that aims to cast a light on *how* deep models make predictions by extracting cross-category visual words and aligning learned visual words with the base deep models' visual attention. LVW streamlines ProtoPNet (Chen et al., 2019) by relaxing its class-specific constraint. Its attention-guided semantic alignment preserves high fidelity with the base deep model on the level of predictive ability and model attention. This fidelity ensures our model loyally interprets the reasoning of the networks, allowing these learned visual words to guide the downstream tasks like model diagnosis. We conducted extensive experiments on six benchmarks to quantitatively our model with recognition accuracy and our proposed IoU coverage metric against three state-of-art prototype-based methods. The experimental results evinced the efficacy and generalization of our LVW method over several benchmarks.

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

# A   APPENDIX

## A.1   DATASET

We include six datasets for performance evaluation. (1) *STL10* (Coates et al., 2011) is a popular image classification benchmark containing 13,000 labeled images from 10 object classes. 5,000 images are used for training while the remaining images are kept for testing. (2) *Oxford 102 Flower* (Nilsback & Zisserman, 2008) is a fine-grained image classification dataset including 102 flower categories. Each class consists of between 40 and 258 images. Based on its original split, we use its training set for training and its validation set for testing. (3) *Oxford-IIIT Pets* (Parkhi et al., 2012) has 37 categories including various breeds of dogs and cats, with roughly 200 images for each class and it is often used for fine-grained image classification tasks. (4) *Stanford Dogs* (Khosla et al., 2011) contains 20,580 images with fine-grained annotation for 120 different dog breeds. The sample images are divided into 12,000 images for training and 8,580 images for testing. (5) *Food-101* dataset (Bossard et al., 2014) is a large-scale fine-grained classification benchmark, which contains 101 food categories with 750 training and 250 test images per category. (6) *AwA2* (Xian et al., 2019) contains 37,322 images in 50 animal categories. We choose this dataset specifically for checking if our method generalized well for non-fine-grained datasets. Notice that all sample images are not cropped, even for those datasets that come with Region of Interest annotation.

## A.2   IMPLEMENTATION DETAILS

We implement our model in PyTorch (Paszke et al., 2019) using one NVIDIA Titan V GPU. For all datasets, we fine-tune the ResNet-34 base model pre-trained on *ImageNet* (Deng et al., 2009) for each dataset, and then Grad-Cam (Selvaraju et al., 2017; 2020) attention map is extracted with the fine-tuned ResNet. All the images are resize to $224{\times}224{\times}3$ except for *STL10*. We keep the original image size for *STL10* as all its images have the same size. Random horizontal flip is applied during training as the only data augmentation procedure. The shape of convolutional output $f(x)$ is $H{=}W{=}7$ with $D{=}128$ channels. The shape of each visual word is set to $H'{=}W'{=}1$. The learning rate of the backbone feature extractor is set as 0.0001, while the learning rate of the other parts is set as 0.003. $\alpha$, $\beta$ and $\gamma$ are set to 0.8, 10 and 0.0001, respectively. The number of visual words $k$ is set to 10 empirically for attention-guided alignment training. All datasets are trained for 200 epochs, and we project visual words after every 10 epochs of training. We use ReLU as the activation function except for the last fully connected layer, in which we use the sigmoid function.

## A.3   ALGORITHMIC PERFORMANCE ON VGG BACKBONE

Our method can also be used for other types of convolutional backbones, such as VGG (Simonyan & Zisserman, 2014), DenseNet (Huang et al., 2017), and potentially ViT (Dosovitskiy et al., 2020). We replace the ResNet-34 backbone with VGG-19. We also trained a base VGG-19 model on the *Oxford-IIIT Pets* (Nilsback & Zisserman, 2008) and obtain the Grad-CAM as the input attention for our method. Here we report the performance of different interpretation methods with the VGG backbone on *Oxford-IIIT Pets* in Table 2. Our method still outperformances all baseline methods in terms of accuracy and IOU coverage.

Table 2: Performance of different interpretative models with VGG-19 backbone on *Oxford-IIIT Pets*

| Method | Oxford-IIIT Pets | |
| --- | --- | --- |
| | Acc | IoU |
| VGG-19 (Simonyan & Zisserman, 2014) | 89.06 | 1.00 |
| ProtoPNet (Chen et al., 2019) | 86.24 | 0.3696 |
| ProtoTree (Nauta et al., 2021) | 50.14 | 0.3305 |
| ProtoPShare (Rymarczyk et al., 2021b) | 87.04 | 0.2722 |
| TesNet (Wang et al., 2021) | 87.53 | 0.5688 |
| Ours w/o Atte. Align. | 87.57 | 0.2321 |
| Ours | **87.93** | **0.6426** |

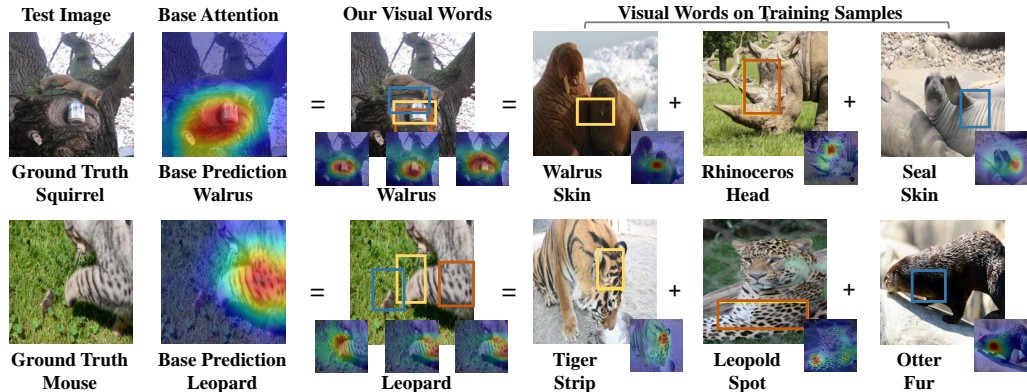

Figure 5: Examples where the base model makes wrong predictions based on incorrect attention. The first column is the test image, and the second column is the base model's attention map and prediction. The third column contains the detected visual words in the sample and the prediction of our model.

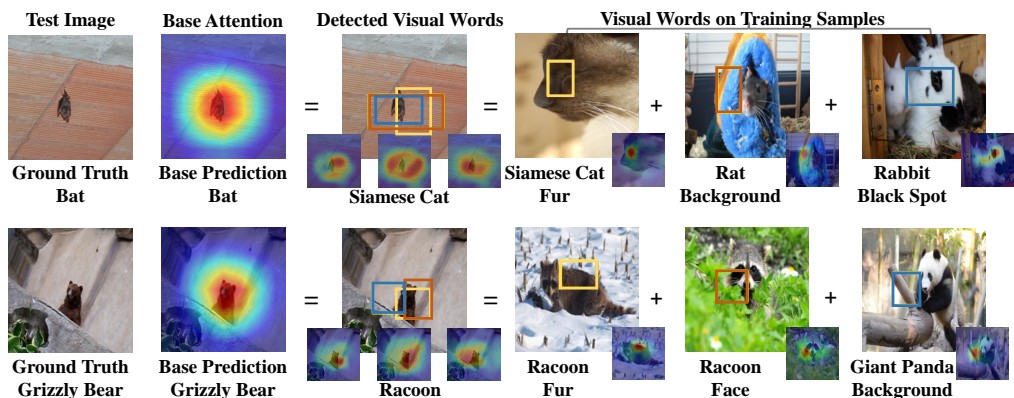

Figure 6: Failure cases where our model makes wrong interpretations based on the correct attention of the base model.

## A.4 FAILURE CASE ANALYSIS

In this section, we share some failure cases for both the base model and our interpretative model to identify the potential limiting factors of our method.

**Failure cases of the base model.** Firstly we show two examples where the base model makes wrong predictions because of incorrect attention in Figure 5. In the upper example, a "squirrel" image is predicted as "walrus" by the base model. The attention map shows that the base model focuses on the foreground soda can and tree stump instead of the squirrel. Guided by this attention, our model finds out that the visual word representing the walrus skin is similar to the texture of the tree, and the visual words representing the rhinoceros head and seal skin are similar to the texture/color of the soda can. The lower example is labeled as a "mouse" even though the salient object is a cat-like animal. Unsurprisingly, the base model predicts this image as a "leopard" based on the foreground object. In this case, our model related the strips and spots of the foreground animal with the tiger's strips and the leopard's black spots. In both failure examples, our visual words provide a reasonable explanation of how the base model makes a wrong prediction, which validates the effectiveness of our visual words in model diagnosis. Notice that the last visual word representing the black fur of the otter finds the mouse despite it being located outside of the base attention. We conjure that the high $l_2$ similarity between these two patches causes this visual word to deviate from the base model's attention.

**Failure cases of our interpretative model.** Next, we will show two failure examples of our model when interpreting the correct predictions of the base model in Figure 6. In both cases, the base model makes the correct predictions and the base model provides reasonable attention maps for the predicted categories. However, the main object only occupies a very small area in each example, which might

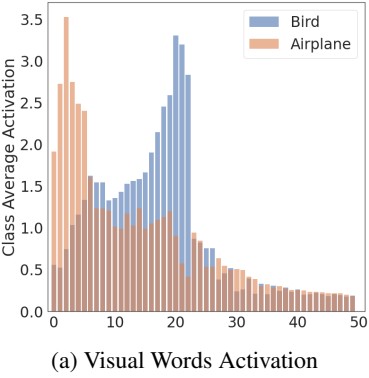 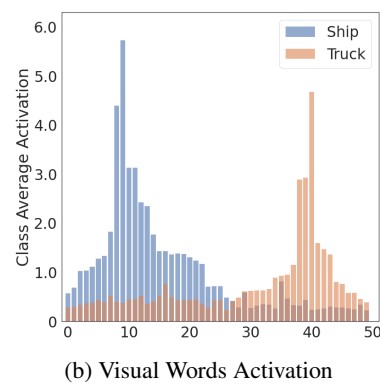

(a) Visual Words Activation
of Similar Categories

(b) Visual Words Activation
of Dissimilar Categories

Figure 7: Visual words exploration. (a) demonstrates the visual word distributions on two similar categories and (b) demonstrates the distributions of two dissimilar categories, where the x-axis denotes the 50 learned visual words.

only contain limited semantic information. A small number of visual words will be enough to extract such information, while other visual words might align with the background noise. For example, in the upper "bat" example, the first visual word finds the similarity between the color of a "siamese cat" and the color of the "bat", while the other two visual words seem only to focus on the background. Similarly, the visual words in the second example fail to connect the small attention area in the test image to the "grizzly bear" category, resulting in an incorrect prediction and explanation.

Based on these examples, it seems that learning the prototypes solely on the $l_2$ similarity guided by the attention map may impair the performance of our method, especially for extreme cases shown in Figure 6. Finding a better similarity measurement for aligning semantics or refining the attention map with other techniques e.g., posture detection, would be interesting extensions for our future work.

## A.5 VISUAL WORDS EXPLORATION

We explore the learnable visual words by inspecting the interrelation between different classes. As we discussed in Section 3.2, the output of visual words learning module $g \circ f(x_i)$ can be viewed as an activation vector representing how strong each visual word is associated with training image $x_i$. By averaging the vectors of all training samples in the category $c$, we acquire a new vector representation of the category $c$ defined by its similarity with the $M$ learned visual words. We choose two pairs of classes, one similar and one dissimilar, and plot each pair's visual words based activation vectors together in Figure 7(b)&(c). As demonstrated, the similar pair of "Bird" and "Airplane" has relatively high activations on at least ten visual words, which suggests these shared visual words' presence in both categories, while those unshared visual words differentiate one class from another. On the other hand, the dissimilar pair "Truck" and "Ship" barely agree on any visual words.

