# OpenReview forum: "Learnable Visual Words for Interpreting Image Recognition Models"
_ICLR.cc/2023/Conference — Submitted to ICLR 2023_

### Official Review · Reviewer_HEcT · 2022-10-24

**Confidence:** 3
**Correctness:** 3
**Technical Novelty And Significance:** 2
**Empirical Novelty And Significance:** Not applicable
**Recommendation:** 5

**Clarity, Quality, Novelty And Reproducibility:**

The paper is well written with analysis and experiments and can be reproduced from the details in the paper.

**Strength And Weaknesses:**

Strengths - The paper tackles the problem of understanding and interpreting  deep neural networks by learning visual words and grounding the words aligned with the attention of the base model. They perform experiments on several benchmarks and present  several analysis.

Weaknesses -

1. The idea about aligning the maps of visual words to the base model attention map will prevent stronger deviations from learning wrong attentions but at the same time also will prevent the model to learn correct disambiguations. How do you ensure that the models learn part speciifc attentions for the learned visual words ?

2. Also what would happen when the base model's attention is completely wrong ?

3. The contributions are very thin compared to previous literature  and the improvements in the IoU metric is obvious since the model's objective is to improve the overlap between attentions. But does this metric correctly state that the model is more interpretable ?

4. It would be worthwhile to share some failure cases of the model to get a more in depth idea of what are the limiting factors.



**Summary Of The Paper:**

In this paper, the authors propose to learn visual words to explain deep neural networks specifically for the task of image recognition. Certain visual words are learned based on the target category and these words are encouraged to be shared between different categories. Moreover, they also include the attention guided semantic alignment that encourages the leaned visual words to focus on the same conceptual regions for prediction. They show results on several benchmark datasets on both accuracy and interpretation fidelity. They also show that their method is generalizable to ground visual words for unseen categories,.

**Summary Of The Review:**

I have some concerns regarding the contributions and the training process adopted to encourage interpretability. It is not very clear if the scores are high because of correct attention map guidance or something else. An in-depth evaluation of learned visual words, failure cases and how the attention maps differ from baseline when these are wrong would be interesting to see.

---

> ### Author Response · Authors · 2022-11-15
> **Response to Reviewer HEcT (2/2)**
>
> **Correctness of the IoU metric.** We would like to point out that our novelty comes from our observations and analyses of the research problem and the philosophy to address the new problem arising from such observations. We propose the IoU metric based on the observation that current accuracy-driven methods tend to explain their decisions with semantic information from the background, which differs from the base model attention for making predictions. We will illustrate the correctness of the IoU metric in two aspects.
>
> *The goal of our IoU metric.* Our observation of current prototype-based methods indicates that accuracy alone cannot guarantee the interpretative model's loyalty toward the base network. The existing methods attempt to achieve as high accuracy as possible, while at the same time, providing some human-understandable examples to explain their decisions. We noticed that some prototypes of ProtoPNet focus on the most salient part, while some focus on the image background border areas. These off-attention prototypes do not affect classification accuracy in general despite the fact that they are not attended by the base network. Therefore, we proposed the IoU metric to calculate the overlap between the attention of the visual words and the attention of the base model. This metric quantitatively measures the loyalty of a model's interpretation toward the base network. A higher IoU coverage means that the visual words focus more on the semantics information that is utilized by the base model, which indicates the model with a higher IoU metric provides a more faithful interpretation of the base model compared with models with lower IoU. We believe the IoU metric correctly reflects the level of interpretability in terms of fidelity to the base model.
>
> *The correspondence between IoU metric and interpretation fidelity.*  The correctness of our IoU metric is validated in Figure 2 with visual demonstrations. For the same test image, our model with the highest IoU coverage discovers various semantics covered by the base model's attention. Specifically, our visual words pinpoint image patches associated with “cheese crumbs,” “tomatoes,” and “green vegetables”, which are all covered by the base model attention. In our opinion, our method provides a faithful interpretation of the base model's reasoning process. On the other hand, the methods with lower IoU values find image areas associated with the edge of the plates or background noise located away from the base model attention, which implies these prototypes do not reflect the reasoning process of the base model with high fidelity.
>
> **Failure cases.** That is a very good suggestion. We will show some failure cases in the updated manuscript and provide further in-depth explorations. We add a section to share some failure cases in the appendix, as we cannot put images in our responses. We invited the reviewer to check Appendix Section A.4 in the updated manuscript for these failure examples.
>
> Again, we appreciate Reviewer HEcT's insightful comments and suggestions. If any questions remain unclear to the reviewer, we are happy to provide more evidence and explanations.

---

> ### Author Response · Authors · 2022-11-15
> **Response to Reviewer HEcT (1/2)**
>
> Thank you for dedicating your time to reviewing our paper with a high level of expertise. The reviewer brings up some interesting questions on this topic and we would like to provide point-to-point responses as follows.
>
> **Disambiguation of the learned visual word.** We adopted the three-stage training schema of ProtoPNet to ensure the disambiguation of visual words. Adding base model attention will not prevent the disambiguation of the learned visual words, as the attention alignment is not intended to disassemble the base attention into pieces for individual visual words. Instead, the base model attention only encourages the visual words to focus more on the same areas attended by the base model instead of background noise. The pushing stage during training will push each visual word to one of the latent feature patches, which ensures all the visual words can be traced back to the individual image parts in the training samples. With the guidance of attention alignment, the image patches associated with our visual words are more likely to overlap with the base model attention. We believe these training image patches still help disambiguate the visual words. Even when two visual words track down to the same part for a test sample, we can differentiate their semantic meanings by referring to the original image patch corresponding to each visual word. For example, if a visual word associated with a tiger's body and another visual word associated with a horse's head both point to the head of a zebra, we can intuitively understand that they represent different semantics: the body strips and the head shape of a zebra, respectively.
>
> **Wrong base model attention.** As we discussed in our motivation, we believe a good interpretation model should be loyal to the base model, i.e., if the base model produces a prediction, right or wrong, with *incorrect* attention, the interpretation model is also expected to explain how the base model makes such a prediction with the information from the same *incorrect* image areas. For instance, the base model might predict an image as a "boat" while attending to the background water instead of the boat object. In this case, a good interpretative model should also be able to identify the color or the texture of the water as the key factors contributing to the "boat" prediction. If this behavior is undesirable for the users, model designers can diagnose and calibrate the model bias based on the interpretation, which is part of our motivation for attention alignment.
>
> We will address Reviewer HEcT's last two questions in the second comment block.

---

### Official Review · Reviewer_Tm8e · 2022-10-25

**Confidence:** 3
**Correctness:** 2
**Technical Novelty And Significance:** 2
**Empirical Novelty And Significance:** 2
**Recommendation:** 3

**Clarity, Quality, Novelty And Reproducibility:**

I think the originality is limited.


**Strength And Weaknesses:**

Strength
1. I think model interpretability is quite an interesting task.
2. The paper is well-written and easy to understand.

Weakness
1. It seems the novelty is limited as it is merely a simplification of the current work ProtoPNet. It simply removes the separation loss. However, the motivation for removing this loss is not well clarified. Besides, as it is a key contribution of this work, but its effectiveness is also not verified in the experiments.
2. It seems that semantic alignment is also used in many model interpretability works. Its originality should also be better clarified.

**Summary Of The Paper:**

This work proposes a learnable visual word model that consists of two modules to predict model prediction behaviors. Concretely, a semantic visual words learning model is proposed to relax the category-specific constraint and thus enable the generic visual words shared across multiple categories. A dual fidelity preservation module is proposed to encourage the learned visual words to focus on the same conceptual regions for prediction.


**Summary Of The Review:**

I think the originality is limited, and the experiments seem not sufficient.

---

> ### Author Response · Authors · 2022-11-15
> **Response to Reviewer Tm8e (2/2)**
>
> **Originality of our semantic alignment.** Yes, we agree with Reviewer Tm8e that the concept of semantics is widely used in interpretability methods. Various methods choose to embody this concept of semantics in different ways. We would like to clarify the differences in the following points.
>
> *Neuron activation as hidden semantics.* Works including [1, 2] aim to make sense of certain hidden neurons/layers as hidden semantics. This kind of interpretation can only find some important input patterns as semantics, whereas our method uses latent visual words and their associated training image patches to visualize semantics.
>
> *Feature attribution as semantics.* Attribution-based interpretative methods [3, 4] identify *which* parts of input images contain the semantics corresponding to the final prediction. Different from these methods, our model adopts the prototype framework from ProtoPNet [5] to link the test image with semantic-related prototypes, providing visual cues on *how* deep models make predictions.
>
> *Prototypes as semantics.* In contrast to ProtoPNet and its extensions [6, 7], our method aligns the semantic information represented by the prototypes (*how*) with the attribution attention map (*which*), leading our visual words to focus on the same image area as the base deep model. To the best of our knowledge, our method is the first interpretative work that bridges the gap between *which* and *how* with semantic alignment, with an intention to increase the fidelity of the interpretation toward the base model.
>
> If any questions remain unclear to the reviewer, we are happy to provide more evidence and explanations.
>
>
> [1] F. Wang, H. Liu, and J. Cheng, “Visualizing deep neural network by alternately image blurring and deblurring,” Neural Networks, vol. 97, 2018.
>
> [2] R. Fong and A. Vedaldi, “Net2vec: Quantifying and explaining how concepts are encoded by filters in deep neural networks,” in The IEEE Conference on Computer Vision and Pattern Recognition, 2018.
>
> [3] Selvaraju, Ramprasaath R., et al. "Grad-cam: Visual explanations from deep networks via gradient-based localization." Proceedings of the IEEE international conference on computer vision. 2017.
>
> [4] Shrikumar, Avanti, Peyton Greenside, and Anshul Kundaje. "Learning important features through propagating activation differences." International conference on machine learning. 2017.
>
> [5] Chaofan Chen, Oscar Li, Daniel Tao, Alina Barnett, Cynthia Rudin, and Jonathan K Su. This looks like that: deep learning for interpretable image recognition. Advances in Neural Information Processing Systems, 2019.
>
> [6] Jiaqi Wang, Huafeng Liu, Xinyue Wang, and Liping Jing. Interpretable image recognition by constructing transparent embedding space. In IEEE International Conference on Computer Vision, 2021.
>
> [7] Meike Nauta, Ron van Bree, and Christin Seifert. Neural prototype trees for interpretable fine-grained image recognition. In IEEE/CVF Conference on Computer Vision and Pattern Recognition, 2021.

---

> ### Author Response · Authors · 2022-11-15
> **Response to Reviewer Tm8e (1/2)**
>
> We greatly appreciate the time and effort you expended in reviewing our paper. We want to provide point-to-point responses as follows.
>
> **Motivation and effectiveness of removing separation loss.** We would like to point out that our novelty comes from our observations and analyses of the research problem and the philosophy to address the new problem arising from such observation, which, in our personal option, constitute the key novelty of a paper. We will explicate our motivation for removing the separation loss and its effectiveness here.
>
> *Motivation of removing separation loss.* Our motivation for removing separation loss is based on the observation that ProtoPNet learns categorical specific prototypes by forcing each prototype to belong to only one category. Ideally, ProtoPNet would only use the prototypes of the "cat" category to interpret the test images predicted as "cat", while all prototypes of other categories should have no effect or even a negative effect for classifying a sample as "cat."  On the contrary, we believe different categories might share similar prototypes, e.g., both cats and dogs have fur, paws, and so on. And prototypes representing the fur of the "dog" category should not be considered as evidence against "cat." Therefore, we hope to learn cross-category prototypes. Since we notice that a simplified version of ProtoPNet can tackle the across-category prototypes, we have no motivation to design more complicated components to achieve this goal.
>
> *Effectiveness of removing separation loss.* Bearing the above motivation in mind, we illustrate the effectiveness of our method in Figure 2 and Figure 4. Figure 2 demonstrates that our method indeed discovers cross-category prototypes that achieve comparable predicting ability with the category-specific methods. Additionally, we verified that the cross-category prototypes can detect similar semantics from images from other classes in Figure 4, even for classes that are **unseen** during training. This validates that removing separation loss brings the benefits of better **generalization** ability in interpreting **unseen** categories. We also demonstrate that our cross-category prototypes are more likely to extract similar semantics from categories that are closely related in the last section of the Appendix. We did not conduct experiments to compare the predicting accuracy between our full model and our model with separation loss as a higher accuracy is not our goal. We kindly request Reviewer Tm8e to be more specific on what additional experimental results the reviewer expects to see for verifying the effectiveness of removing the separation loss.
>
> We will address Reviewer Tm8e's last question in the second comment block.

---

### Official Review · Reviewer_zev4 · 2022-10-27

**Confidence:** 2
**Correctness:** 1
**Technical Novelty And Significance:** 2
**Empirical Novelty And Significance:** 2
**Recommendation:** 5

**Clarity, Quality, Novelty And Reproducibility:**

- The paper is clearly written. and seems reproducible.

**Strength And Weaknesses:**

[Strength]

- The paper is well-written and clear.

- Experiments results show the proposed method is better than previous methods.

[Weakness]

- My major concern with the proposed method is its novelty. The proposed LVW basically follows ProtoPNet's objective and removes the separation loss. The ProtoPNet's objective is also learnable.

- The metric of IoU is also problematic, it's unfair to compare to other methods using this metric since the proposed method trained with l2 normalization loss.

- What is the performance with separation loss? It seems there is no such number in the paper.

**Summary Of The Paper:**

This paper proposed a learnable visual word method to interpret the model prediction behaviors. The authors proposed 2 new modules: semantic visual word learning and dual fidelity preservation. The semantic visual word removes the separation loss from the protopnet, and the dual fidelity preservation adds attention alignment with the grad-cam outputs. Experiments show the proposed method achieves good performance on the metric.

**Summary Of The Review:**

This paper provides two improvements over the prior ProtoPNet's architecture: 1: remove the separation loss; 2: add attention-guided alignments. The proposed IoU metric is a little bit concerning since the model directly optimizes for this target compared to other methods.

---

> ### Author Response · Authors · 2022-11-15
> **Response to Reviewer zev4 (2/2)**
>
>
> **Fairness of IoU metric.** In our opinion, our IoU metric is fair for the intended purpose, and we would like to elucidate from two aspects.
>
> *The purpose of the IoU metric is irrelevant to fairness.* The IoU metric is a quantitative measurement to validate if our method fulfills the motivation. As we discussed, our goal is to loyally interpret the base model, which is different from the motivation of existing ProtoPNet-like methods. To that end, we introduced the IoU metric, which calculates the overlap between the attention of the visual words and the attention of the base model to quantitatively measure such loyalty. In this sense, evaluating with IoU metric does not introduce unfairness. Similar to fairness learning, performance should be evaluated by utility and fairness. Here we evaluate the interpretative model by both recognition accuracy and IoU. Our results validate that our model provides a more faithful explanation for the base model, which is one of our goals. To our best knowledge, there are no prototype-based studies targeting this same goal. So we put the related methods in the same research problem for comparisons as reference. We believe it is better to involve competitive methods than only showing the visual example.
>
> *Our method does not need extra inputs.* Our method indeed is trained against the base model attention with l2 loss, but we do not involve extra information as supervision besides the class labels. The attention maps come from the base model and only require the sample images and the class labels, just like the baseline models in our comparison. In our method, the base model attention provides a new view of the training data without additional inputs, much like an augmentation process. For example, nearly all current works in contrastive learning contain an essential data augmentation module for creating new views of the training images. In our opinion, our model utilized the original inputs to pursue the research goal rooted in our motivation without additional supervision, so we don't consider it an unfair comparison.
>
> **Performance with separation loss.** The orignal ProtoPNet contains separation loss in its overall objective. We invite the Reviewer zev4 to take a closer look at Table 1, which provides the performance comparison between ProtoPNet with and without separation loss. Specifically, our model without attention alignment, whose performance is listed in the second last row as "*Ours w/o Atte. Align.*", is essentially the same model as ProtoPNet without separation loss. The performance difference between this model and the original ProtoPNet can help evaluate the effect of separation loss. The results illustrate that the separation loss does not affect the final accuracy or IOU in any deterministic way. We will make this clearer in our manuscript.
>
> If any questions remains unclear to the reviewer, we are happy to provide more evidence and explanations.
>
>
> [1] Selvaraju, Ramprasaath R., et al. "Grad-cam: Visual explanations from deep networks via gradient-based localization." Proceedings of the IEEE international conference on computer vision. 2017.
>
> [2] Shrikumar, Avanti, Peyton Greenside, and Anshul Kundaje. "Learning important features through propagating activation differences." International conference on machine learning. 2017.
>
> [3] S. Bach, A. Binder, G. Montavon, F. Klauschen, K.-R. Muller, and W. Samek, “On pixel-wise explanations for non-linear classifier
> decisions by layer-wise relevance propagation,” PloS one,  2015.
>
> [4]  M. Sundararajan, A. Taly, and Q. Yan, “Axiomatic attribution for deep networks,” in Proceedings of the 34th International Conference on Machine Learning, 2017.

---

> ### Author Response · Authors · 2022-11-15
> **Response to Reviewer zev4 (1/2)**
>
> We greatly appreciate the time and effort you expended in reviewing our paper. We want to provide point-to-point responses as follows.
>
> **Novelty of our paper.**  We would like to point out that our novelty comes from our observations and analyses of the research problem and the philosophy to address the new problem arising from such observations, which, in our personal option, constitute the key novelty of a paper.
>
> *Cross-category visual words.* Our first observation is that ProtoPNet learns categorical specific prototypes by forcing each prototype to belong to only one category. Ideally, ProtoPNet would only use the prototypes of the "cat" category to interpret the test images predicted as "cat", while all prototypes of other categories should have no effect or a negative effect for classifying a sample as "cat."  On the contrary, we believe different categories might share similar prototypes, e.g., both cats and dogs have fur, paws, and so on. We believe these prototypes of the "dog" category should not be considered as evidence against "cat," therefore we expect to learn cross-category prototypes. Besides, the interpretability of category-specific prototypes is limited to the existing categories in the training data, while our cross-category visual words provide additional capacity for interpreting **unseen** categories, which brings the benefits of better **generalization** ability for our method. As shown in Figure. 4, our method explains the new categories with visual words corresponding to the existing categories during training. Our cross-visual words interpret a "*liger*" as the combination of the tiger's body strips, the lion's face, and the lion's fierce hair, even though the model has never seen the “*liger*” category. Since we notice that a simplified version of ProtoPNet can tackle the across-category prototypes, we have no incentive to design more complicated components to achieve this goal.
>
> *The fidelity towards the base network.* Secondly, we notice that the similar accuracy between the base model and the interpretation model does not ensure the interpretation faithfully untangles the reasoning process of the base model. We believe the fidelity to the base model is no less important than the predicting accuracy for an interpretative model. For this reason, we propose attention alignment to guide the focus of our learnable visual words. This motivation is validated in Figure 2, where some prototypes of ProtoPNet focus on the most salient part, while some focus on the image border. As demonstrated in previous visual attention interpretative models [1,2,3,4], the backbone method is very unlikely to classify a Greek salad as "greek salad" based on the edge of a plate. But in general, these prototypes together can still predict similarly to the base model, which means only focusing on a higher accuracy does not guarantee the quality of the interpretation for the base model. Therefore, we add the base model attention as a key factor into consideration for fidelity and propose dual fidelity preservation to encourage the learned visual words to preserve high fidelity with the base model in terms of both prediction and model attention.
>
> In our opinion, the motivation and the philosophy of solving the challenge brought by this motivation, compose the novelty of our paper.
>
> We will address Reviewer zev4's other questions in the second comment block.

---

### Decision · Program_Chairs · 2023-01-20

**Decision:**

Reject

**Justification For Why Not Higher Score:**

limited novelty and result significance insufficient

**Justification For Why Not Lower Score:**

NA

**Metareview: Summary, Strengths And Weaknesses:**

The paper proposes a learnable visual word method to interpret the model prediction behaviors with a semantic visual words learning model to relax the category-specific constraint and thus enable the generic visual words shared across multiple categories, and a dual fidelity preservation module to encourage the learned visual words to focus on the same conceptual regions for prediction.
Overall the contribution is rather limited in terms of novelty (over similar work ProtoPNet) and result improvements.


**Summary Of Ac-Reviewer Meeting:**

Not needed as no reviewer accepts the paper